# FakeFlow: Fake News Detection by Modeling the Flow of Affective Information

## Reproducibility Summary

**Scope of Reproducibility**

Unlike in short news texts, authors of longer articles stir the readers' attention by means of emotional appeals that arouse their feelings. To capture this, Bilal et al. (2021) propose in their paper to model the flow of affective information in fake news articles using a neural architecture. The authors claim to introduce a model, FakeFlow for learning flow of affective information in fake news articles that outperforms the state-of-the-art methods for this task.

**Methodology**

To reproduce the results of the paper, the main experiments are reproduced. We implement FakeFlow from scratch in TensorFlow and rely on the description provided in the original paper, and referred to the authors' code for only specific implementation detail. We evaluate FakeFlow for MultiSourceFake data requiring 4 hours per training run on Google Colab Pro. All code used is publicly available at Github https://github.com/asifajunaidahmad/ReproducibilityChallenge21.

**Results**

The FakeFlow model obtained 96% accuracy at 6th epoch and stopped due to early stopping function. We called back the early stopping function and run the model upto 50 epochs as mentioned in paper. The model performance increased 1% only but loss started to increase as the number of epochs increased.

Going beyond the original paper, we created confusion matrix to further investigate the reason of wrong labels classification.

**What was easy**

The authors provide code for most of the experiments presented in the paper on Github. The code available with this paper for FakeFlow implementations was easy to run and allowed us to verify the correctness of our re-implementation. A thorough and clear description of the proposed algorithm was provided in the paper without any obvious errors or confusing exposition.

**What was difficult**

It took considerable time and effort to understand that code works only on specific versions of the Libraries, mentioned in the Readme file. These versions of the Libraries are outdated and I have to reinstall older versions of most of the Libraries to work. Moreover, to run the complete dataset of the paper, additionally GPU resources like Google Colab Pro were required. Purchased additional resources to run 50 epochs of the code.Morerover, no information about data format was provided in the paper. We figured out after lot of effort that data set was working in particular format only.

**Communication with original authors**

Communication with the authors was attempted but could not be established.

Submitted to ML Reproducibility Challenge 2020. Do not distribute.

# 1 Introduction

In fake news articles, the authors exploit the length of the news to conceal their fabricated story. The flow of information has been investigated for different tasks: Reagan et al. (2016) studied the emotional arcs in stories in order to understand complex emotional trajectories; Unlike previous works (Rashkin et al., 2017; Shu et al., 2018; Castelo et al., 2019; Ghanem et al., 2020) that discarded the chronological order of events in news articles, in this work authors propose a model that takes into account the affective changes in texts to detect fake news.

The paper " FakeFlow: Fake News Detection by Modeling the Flow of Affective Information " Bilal Ghanem et al. [2021] hypothesize that fake news has a different distribution of affective information across the text compared to real news. Therefore, modeling the flow of such information can help discriminating fake from real news. The authors propose FakeFlow, a model that aims to detect fake news articles by taking into account the flow of affective information.

As a part of the ML Reproducibility Challenge, I tried to replicate FakeFlow model from scratch and investigate if the model detects fake news articles by taking into account the flow of affective information.

# 2 Scope of reproducibility

In this review, the work of the proposed FakeFlow model by Bilal et al. is reproduced and examined. The aim is to reproduce the results obtained by the authors and to investigate the claims made in the paper. The claims made can be seen below. Each claim will be examined in a corresponding subsection in section

This section roughly tells a reader what to expect in the rest of the report. Clearly itemize the claims you are testing:

- FakeFlow model detects fake news articles by taking into account the flow of affective information
- FakeFlow outperforms BERT in detecting fake news articles due to difference of input length in these models
- MultiSource Fake data validate the correctness and effectiveness of the proposed FakeFlow method and demonstrate its practical advantages over other existing methods.

# 3 Methodology

Most of the original source code was available and used to test the reproducibility of the paper, which can be found in the corresponding GitHub repository of the aper. This repository itself contained code from the repository of Fakeflow.

The FakeFlow implementations were used largely as is, furthermore, some small optimisations were made to make certain functions more efficient. For reproducing the results the models were trained on a Google Colab Pro. The code for creating the visualisations was not included in the repository and also no code for comparison with other State-of -the-Art-Method is included. The implementation was made using the specific versions of tensorflow and NumPy etc. libraries with Python.

## 3.1 Model descriptions

The proposed FakeFlow model has two main modules: The first module uses a Convolutional Neural Network (CNN) to extract topic-based information from articles. The second module models the flow of the affective information within the articles via Bidirectional Gated Recurrent Units (Bi-GRUs).

We did not change or modify structure or any other detail of the model and used it just as mentioned in the paper to see how it performs on MultiSourceFake data. The model gave 96% accuracy at 6th epoch and stopped due to early stopping function. We called back the early stopping function and run the model upto 50 epochs to see how model perform. The accuracy increase 1 percent but valid loss started to increase as number of epochs increased.

## 3.2 Datasets

We conduct our experiment on MultiSouceFake dataset; this dataset is created after relying on different resources for creating the training and test portions of the dataset, so as to provide a challenging benchmark. The training parts include 9,708 articles while test part includes 1689 fake news articles. I use a train/val/test split.

Given an input document, the FakeFlow model first divides it into N segments. Then it uses both word embeddings and other affective features such as emotions, hyperbolic words, etc. in a way to catch the flow of emotions in the document. The model learns to pay attention to the flow of affective information throughout the document, in order to detect whether it is fake or real.

In the paper, results are reported on a dataset splitting the articles' text into N segments and set the maximum length of segments to 800 words, applying zero padding to the ones shorter than 800 words. During experiment we noticed that model work only in specific format of the data with columns in specific order otherwise code gives error. Moreover, Ids column of the data wok if ids are shuffled, incase of sequence code was giving errors.

## 3.3 Hyperparameters

The authors of the original paper mention specific values for some of the hyperparameters. However, for other hyperparameters only a range is provided without a clear indication of what values were used for each evaluation

Dropout: random selection in the range [0.1, 0.6]

Dense layers: [8, 16, 32, 64, 128]

Activation functions: [selu, relu, tanh, elu]

CNN filters' sizes: [(2, 3, 4), (3, 4, 5), (4, 5, 6), (3, 5), (2, (4,), (5,), (3, 5, 7), (3, 6)]

Numbers of CNN filters: [4, 8, 16, 32, 64, 128]

Pooling size: [2, 3],

GRU units: [8, 16, 32, 64, 128],

Optimization function: [adam, adadelta, rm- sprop, sgd],

For the early stopping, the 'patience' parameter to set 4 and the number of epochs is 50.

These same hyperparameters were used for our experiments.

## 3.4 Experimental setup and code

Accuracy,Percsion, Recall and F1 Score and Confusion Matrix were used to evaluate the Performance of the Model.

## 3.5 Computational requirements

The experiment was run on Google Colab Pro, training of FakeFlow took approximately 4 hours per training.

# 4 Results

When implementing the full architecture, we found that, despite having all parameters at hand, it would have been helpful to have access to more information concerning the data preprocessing, to achieve performance similar to that reported by the authors.

From our perspective, we cannot tell whether or not better PYTHON programming skills would have been beneficial and if someone with more experience with the multi-head attention would have been able to understand and implement the method right away. In any case, the authors certainly developed a coherent methodology and, by providing the corresponding code alongside with the paper, have ensured that all interested parties can clearly follow their ideas.

The model achieve accuracy as mentioned in the paper, hence proving the author's claim that FakeFlow model detects fake news articles by taking into account the flow of affective information. Secondly, the data used for the FakeFlow model consists of articles, which has longer input length than the data used to train the BERT model. Although no code is provided by author for testing of MultiSourceFake data on BERT model.Since BERT is trained on different type of data we can say that BERT would not have performed better that FakeFlow on MultiSourceFake data. Lastly, MultiSource Fake data validate the correctness and effectiveness of the proposed FakeFlow method

and demonstrate its practical advantages over other existing methods as it achieve the accuracy of 96%. We also created the confusion matrix to further understand the classification of the data by model as shown in Table1 below.

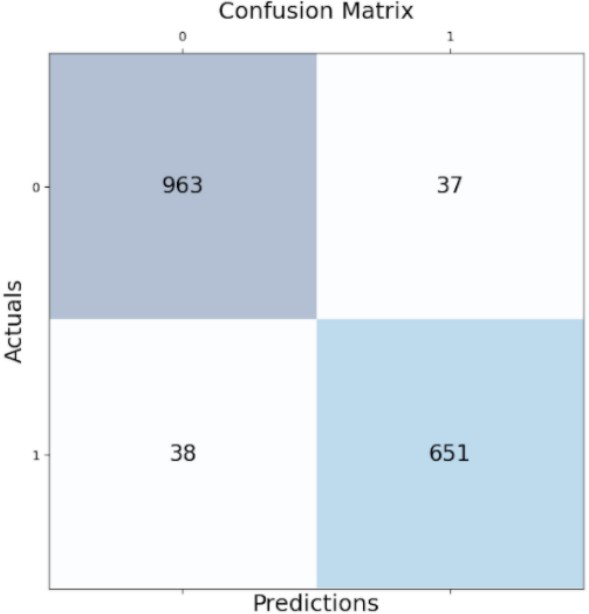

Table1: Confusion Matrix of the FakeFlow Model

So, it can be said that the well-documented code and clean GitHub repository contributed strongly to our understanding of the method and helped us answer most of our comprehension question. Since this all is publicly  available, a reproduction of the presented method based on that implementation is possible.

### 4.1   Results reproducing original paper

The model achieve accuracy as mentioned in the paper, hence proving the author's claim that FakeFlow model detects fake news articles by taking into account the flow of affective information. Secondly, the data used for the FakeFlow model consists of articles, which has longer input length than the data used to train the BERT model.

Although no code is provided by author for testing of MultiSourceFake data on BERT model.Since BERT is trained on different type of data we can say that BERT would not have performed better that FakeFlow on MultiSourceFake data. Lastly, MultiSource Fake data validate the correctness and effectiveness of the proposed FakeFlow method and demonstrate its practical advantages over other existing methods as it achieve the accuracy of 96%. We also created the confusion matrix to further understand the classification of the data by model as shown in Table1 below.

So, it can be said that the well-documented code and clean GitHub repository contributed strongly to our understanding of the method and helped us answer most of our comprehension question. Since this all is publicly  available, a reproduction of the presented method based on that implementation is possible.

### 4.2   Results beyond original paper

Creation of confusion matrix to better understand the accuracy and find how many articles the model is labeling as False positive and False Negative as mentioned in the Table1.We can see that 38 articles are classified as False negative and 37 articles are classified as false positive. By further investigating reason of wrong classification of these particular sample articles, can further improve the model performance.

## 5   Discussion

Evaluated on articles dataset, the central claims of Bilal et al. [2021] hold true. Firstly the accuracy score that we obtained after training the model on MultiSource Fake data are very similar to the ones reported. Secondly, the confusion

matrix seems to support the claim that the FakeFlow method outperforms other state of the art methods due to longer text.

Lastly, MultiSource Fake data validates the correctness and effectiveness of the proposed FakeFlow as accuracy reaches to 96We tried in our experiment to improve the performance of the model by increasing number of epochs, however, loss started getting worse after 6 epochs, where author applied early stopping. The strength of our approach was that we were generally faithful to the original implementation, using largely the same code, which we examined thoroughly. Therefore, the chance of implementation differences with the original code is very small.

A weakness of our approach was that we did not do any work to examine the other baseline models on other datasets mentioned in paper apart from MultiSource Fake, meaning the generalizability of the model remains an open question. Furthermore, due to the high volume and number of datasets to train and test all models mentioned in the paper, which would took a long time given the fact that training of a single model took approximately 4 hours. For this reason, experimentation done with models, datasets and hyperparameters was limited. The experiments mentioned in the paper were not replicated for similar reasons. Overall, the authors provided a model, which outperforms the previously best method for this problem in a quantifiable measure. Lastly, the model is theoretically well motivated.

Despite these strengths of the original paper, some improvements could be made to further substantiate the claims made in the paper. The samples which models classified, as false positive or false negative could be further investigated to find out the reasons of misclassifications. This will help to make improvements and find tune the model.

## 5.1 What was easy

The code was well organized into separate files for e.g., the FakeFlow model, Features or data, making it easy to quickly find specific parts of the code when needed. Additionally, the original paper is quite complete, straightforward to follow, and lacked any major errors.

## 5.2 What was difficult

There were difficulties in replicating some parts of the paper. The older and specific versions of the libraries were difficult task to uninstall current versions and reinstall particular version to code work.

## 5.3 Communication with original authors

Communication with the authors was attempted but could not be established.

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
