# OpenReview forum: "FakeFlow: Fake News Detection by Modeling the Flow of Affective Information"
_ML_Reproducibility_Challenge/2021/Fall — Reject_

### Official Review · Reviewer_hnMi · 2022-02-25
**Rerunning a single experiment using a fake reimplementation**

**Rating:** 2
**Confidence:** 4

**Review:**

The report clearly summarizes the problem statement of the original paper "FakeFlow: Fake News Detection by Modeling the Flow of Affective Information": detecting fake news articles by modeling the flow of affective information. The authors tried to contact the original authors, but did not succeed.

The authors claim to have reimplemented the software from scratch, but the source code referenced in the report is that from the original paper with only small modifications. This statement is misleading and has to be corrected.

The software was applied to the MultiSourceFake data set and could be used to reproduce that result reported in the original paper. As some hyperparameter values were specified only as ranges in the original paper, these were optimized and their final values provided in the report. The authors did not perform any ablation study. Other models and data sets were not examined. There were no explicit recommendations for better reproducibility.

The report describes the reproducibility experiment clearly. I would have expected more experiments using other data sets or with modifications of the algorithm. And I recommend to replace the image for the confusion matrix by an actual table.

---

### Official Review · Reviewer_Ayq7 · 2022-03-06
**Good work**

**Rating:** 8
**Confidence:** 4

**Review:**

Though communication with the original authors was not established, the author was able to reimplement the work using codes provided and  the reinstallation of newer versions of the needed libraries. The author went beyond the original paper by adding confusion matrix to further investigate the issue of wrong labels classification.

The authors should pay attention to the correctness of words used, so that the intended message can be effectively passed across to the readers..

---

### Official Review · Reviewer_ExyQ · 2022-03-18
**Review: Fake news detection by flow of information-FakeFlow**

**Rating:** 8
**Confidence:** 4

**Review:**

This work focused on detection of fake information in article of moderate length considering the information flow. The authors created a fake news dataset from several sources, and named MultiSourceFake. The work came up with deep networks involving CNN, bi-directional GRU layers for the task of identification. The work first embeds the words to vectors using efficient embedding approach. Next, the work extracted emotions, sentiment, morality and related features from the topic to properly characterize the flow information. The work performed comparative assessment with various recent approaches, on multiple datasets for proper analysis. The authors analyzed the emotions portrayed by the article in the starting and ending of the article, and observed that fake news generally tend to evoke strong emotions from the readers in the starting phase. Hence, the readers experience fear, sadness, and surprise emotions at the beginning of the article. However, the ending of the article generally evokes emotions like joy and anticipation rather than the negative emotions. However, this characteristic may not be universal, and real-world fake news may include some other characteristic. However, the finding is important and can even help the readers in general identify fake news manually with considerable accuracy.
The opensource links for the implementation provided by the authors are easy to understand. It includes proper documentation which can enable the researchers to implement the code according to the authors. The repository codes include provisions for utilizing pre-trained models, extraction of features. The authors have included some sampled data for their MultiSourceFake dataset, which seems an excellent collection. We found out that the results are reproducible and easy to implement.

---

### Meta-Review · Area_Chair_7Pab · 2022-04-08

**Recommendation:** Reject
**Confidence:** 4

**Metareview:**

The paper fails to completely reproduce the experiments from the original paper. For example from the original paper table 2 results are partially reproduced, and table 3 and figures 3, 4, and 6 results are missing.   While some of the results which are presented are interesting, I agree with reviewer hnMi that there should be a more detailed experimental analysis and some recommendations for better reproducibility. The paper presentation is poor and can be significantly improved.

---

### Decision · Program_Chairs · 2022-04-09

Reject